# A Green Approach Based on Micro-X-ray Fluorescence for Arsenic, Micro- and Macronutrients Detection in *Pteris vittata*

Giuseppe Capobianco [1], Giuseppe Bonifazi [1], Silvia Serranti [1], Rosita Marabottini [2], Maria Luisa Antenozio [3], Maura Cardarelli [3], Patrizia Brunetti [3,*] and Silvia Rita Stazi [4]

[1] Department of Chemical Engineering, Materials and Environment, Sapienza University of Rome, Via Eudossiana 18, 00184 Rome, Italy; giuseppe.capobianco@uniroma1.it (G.C.); giuseppe.bonifazi@uniroma1.it (G.B.); silvia.serranti@uniroma1.it (S.S.)
[2] Department for Innovation in Biological, Agri-Food and Forestry Systems (DIBAF), University of Viterbo, Via San Camillo de Lellis snc, 01100 Viterbo, Italy; marabottini@unitus.it
[3] IBPM-CNR c/o Dip. di Biologia e Biotecnologie, Sapienza Università di Roma, Piazzale Aldo Moro, 00185 Roma, Italy; marialuisa.antenozio@uniroma1.it (M.L.A.); maura.cardarelli@uniroma1.it (M.C.)
[4] Dipartimento di Scienze Chimiche, Farmaceutiche ed Agrarie (DOCPAS), Università di Ferrara, Via Borsari 46, 44121 Ferrara, Italy; silviarita.stazi@unife.it
* Correspondence: patrizia.brunetti@uniroma1.it; Tel.: +39-064-991-239-22

**Abstract:** In this study, benchtop micro-X-ray fluorescence spectrometry (μXRF) was evaluated as a green and cost-effective multielemental analytical technique for *P. vittata*. Here, we compare the arsenic (As) content values obtained from the same samples by μXRF and inductively coupled plasma-optical emissions spectrometry (ICP–OES). To obtain samples with different As concentrations, fronds at different growth time points were collected from *P. vittata* plants grown on two natural As-rich soils with either high or moderate As (750 and 58 mg/kg). Dried samples were evaluated using multielement-μXRF analysis and processed by PCA. The same samples were then analysed for multielement concentrations by ICP–OES. We show that As concentrations detected by ICP–OES, ranging from 0 to 3300 mg/kg, were comparable to those obtained by μXRF. Similar reliability was obtained for micro- and macronutrient concentrations. A positive correlation between As and potassium (K) contents and a negative correlation between As and iron (Fe), calcium (Ca) and manganese (Mn) contents were found at both high and moderate As. In conclusion, we demonstrate that this methodological approach based on μXRF analysis is suitable for monitoring the As and element contents in dried plant tissues without any chemical treatment of samples and that changes in most nutrient concentrations can be strictly related to the As content in plant tissue.

**Keywords:** *Pteris vittata*; plant response to arsenic; arsenic monitoring; μXRF; multielemental analysis; ICP—OES; PCA

## 1. Introduction

The hyperaccumulator *Pteris vittata* (*P. vittata*) is particularly effective for remediation of arsenic (As)-polluted environments due to its ability to accumulate As in its fronds [1]. *P. vittata* is capable of taking up arsenate (As$^V$) and arsenite (As$^{III}$) via the plant roots and translocating them through the xylem to the aerial parts (i.e., fronds, [2]). This phytoremediation process, named phytoextraction, is a sustainable and cost-effective approach to managing As-contaminated soil and water and to generating large amounts of contaminated biomass that can be used for As valorisation [3]. Despite the promising potential of phytoremediation, the lack of a fast, non-invasive technique for monitoring and measuring the As concentration in ferns hampers the extensive use of this technology.

The laboratory methods preferred for As measurement require pretreatment, either with acidic extraction or acidic oxidation digestion of the environmental sample [4]. Arsenic in acid solution is then measured using any one of several analytical methods, such as

atomic fluorescence spectroscopy (AFS) [5], graphite furnace atomic absorption (GFAA), hydride generation atomic absorption spectroscopy (HGAAS), inductively coupled plasma–atomic emission spectrometry (ICP–AES), inductively coupled plasma–optical emission spectroscopy (ICP–OES) and inductively coupled plasma–mass spectrometry (ICP–MS). These devices are expensive to operate and sustain, bulky and also require fully staffed laboratories for their maintenance and operation [4]. In addition, these techniques require sample pretreatment with hazardous chemical reagents or solvents for analysis. Direct measurement of the elemental composition of samples through X-ray fluorescence (XRF) does not require chemical sample treatment and can be conducted without creating waste [6]. In more detail, X-ray fluorescence for plant tissue analysis offers a very fast analytical method that involves little sample preparation, besides grinding, and without the need for acid digestion.

There are many reports describing the use of μXRF in the elemental composition analysis of solid samples characterised by high measurement accuracy. μXRF constitutes a convenient tool for the analysis of major elements in organic samples [7–10]. However, this level of sensitivity may be acceptable for sample screening or site surveys, as a large number of relatively inexpensive screening results can be obtained in a short period of time [11]. Recent studies pointed out that μXRF can be useful in environmental analysis for the evaluation of trace metal ions in different materials, such as soil and particles, plants, vertebrates and invertebrates, in addition to various biomass surfaces [12,13].

Other studies have proposed the use of μXRF for heavy metal quantification in plant tissue. Chuparina et al. [14] applied XRF to determine concentrations of heavy metals, i.e., iron (Fe), titanium (Ti), manganese (Mn), chromium (Cr), copper (Cu), nickel (Ni), zinc (Zn), strontium (Sr) and barium (Ba), in the medicinal plant *Hemerocallis minor*, and the XRF data were validated by means of a linear calibration curve obtained using national reference certified plant materials. Indeed, the use of μXRF to analyse the content of lead (Pb) and other heavy metals in plant tissues was suggested by Gallardo et al. [15]. Recently, Byers et al. [16] used Wavelength Dispersive X-ray Fuorescence WD-XRF and/or portable Energy Dispersive X-ray Fluorescence ED-XRF spectroscopy to quantify Pb and other heavy metals in dried and fresh edible plant tissues, confirming the accuracy of XRF by comparison with ICP–MS analysis, but only Coetzee et al. [17] has used an XRF-based technology in comparison with ICP–AES to measure the content of As and other elements in grass samples. By using this technology, the presence and distribution of As have been evaluated [18,19]. However, to date, there are no data on the use of a μXRF-based green methodological approach for monitoring the amount of As and variations in the micro- and macronutrient contents in *P. vittata* that is validated by ICP methodologies. The attractiveness of this analysis for the evaluation of heavy metals in plants is mainly related to the possibility of evaluating macro- and micronutrient changes in response to heavy metals and of monitoring the phytoextraction process efficiency. In this context, the development of sustainable analytical methods [20,21] is an important challenge with respect to the environment by reducing the use of concentrated mineral acids (e.g., nitric acid) required for sample digestion, technical gases and the energy required for operation, given the use of "energy-eating" analytical instruments (e.g., plasma spectrometers) [22]. Moreover, techniques such as μXRF analysis allow for fast and non-destructive analysis and therefore repeated measurements on the same samples. In addition, μXRF facilitates analysis on a more representative quantity of samples compared to the stated classical analytical approaches. However, the analytical performance is sensitive to soil moisture, with a signal loss of 37% recorded for As at 20 wt% soil moisture relative to dry soil with an XRF device [23,24]. For this reason, this study proposes a green, cost-effective and fast methodological approach to evaluate As content and perform multielemental analysis on untreated dried plant samples using benchtop μXRF instrumentation. For this purpose, fronds of ferns grown at different times on two different naturally As-rich soils were dried, ground and analysed by μXRF. Evaluation of the accuracy of this method was performed by comparing μXRF results with those obtained by ICP–OES analysis, and the results

were then validated using a chemometric approach. In addition, principal component analysis (PCA) was used to evaluate the correlation between the As content and macro- and micronutrient contents in *P. vittata* fronds. Furthermore, this technology may be used in the future to evaluate the efficiency of phytoextraction.

## 2. Materials and Methods

### 2.1. Plant Growth and Pinna Powder Preparation

The propagation and growth of ferns were performed in the greenhouse under controlled conditions as previously described [19,25]. Six-month-old ferns with 7–8 fronds and about 30 cm tall were transferred in pots containing soils naturally rich in As, with two different As concentrations: Soil 1 (S1) and Soil 2 (S2), with average As concentrations of 58 mg/kg and 750 mg/kg, respectively (Figure 1).

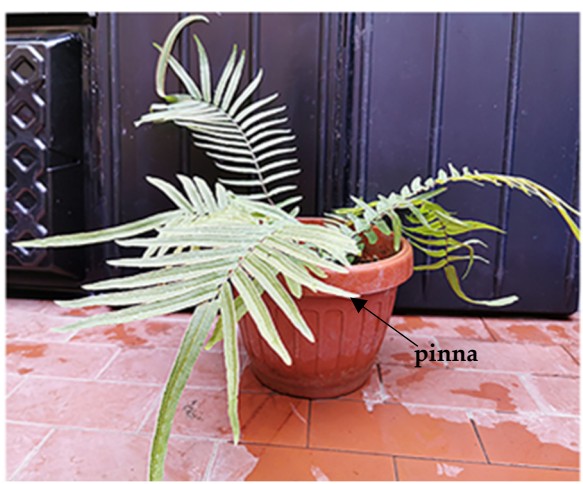

**Figure 1.** A *P. vittata* plant grown in naturally As-rich soil under greenhouse conditions. The arrow indicates the pinna collected for preparing pinna powder.

Samples of all fronds were collected from plants grown in S1 (58 mg/kg DW, Dried Weight), and S2 soils (750 mg/kg DW) at 0, 30 and 60 days, and 0, 30 and 45 days, respectively. In detail, a pinna for each frond was collected from 4 different plants, dried for 48 h at 37 °C and then ground in a mortar into a fine powder (Figure 2).

**A** Collected pinnae    **B** Dried pinnae    **C** Pinna powder

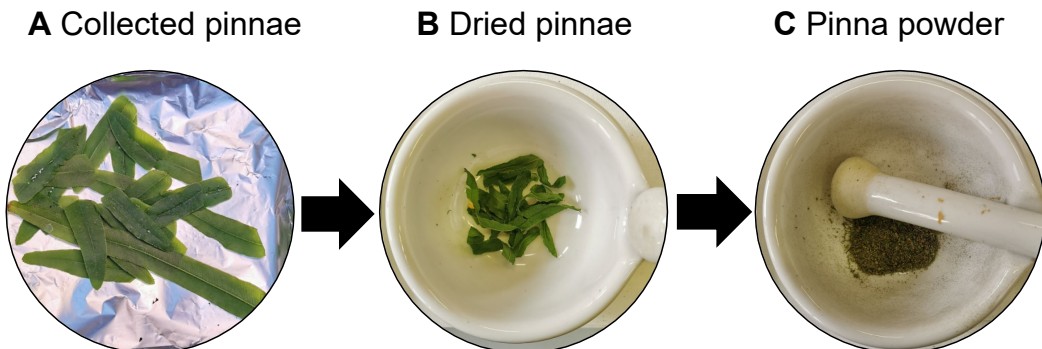

**Figure 2.** Workflow of "pinna powder" preparation. A pinna for each frond was collected from 4 ferns (**A**), dried at 37 °C for 48 h (**B**), and ground in a mortar (pinna powder, **C**).

### 2.2. The µXRF Device

A µXRF benchtop spectrometer Bruker® (Berlin, Germany) M4 Tornado, equipped with an Rh X-ray tube with polycapillary optics and an XFlash detector, providing an energy resolution of better than 145 eV, was utilized to perform the analysis. The polycapillary optic allows the focusing of tube radiation on a very small spot size (~30 µm). Spectrum energy calibration was performed daily, before each analysis batch, by using zirconium

(Zr) metal (Bruker calibration standard). The sample chamber was evacuated to 25 mbar, and, therefore, light elements could be measured.

### 2.3. μXRF Analysis on Dried Pinna Powder

In order to avoid powder dispersion inside the chamber, each sample was divided into 4 stubs and covered by a thin layer (0.01 mm) of polyethylene [18]. All the analyses adopted constant exciting energies (Rhodium, Rh microfocus source 50 kV/400 μA). The element quantification was performed utilizing the Bruker Esprit software 1.2 using the fundamental parameter quantification method (FP) [26]. FP is based on the use of the theoretical relationship between X-ray fluorescence and material composition as determined by Sherman (1955) [27]. The factory-calibrated quantification method of the μXRF device uses this fundamental principle with a calibration based on a Bruker reference standard. Furthermore, the implementation of FP algorithms allows more accurate quantitative analyses within complex matrices like plant materials [28]. The correction factor was based on the proportional ratio between the mean spectra of μXRF powder samples and those obtained from the analysis of the same samples carried out by ICP-OES, expressed in mg/kg. This practice has already been applied in many fields as μXRF quantitative analysis methods [8,29]. Thirty-five punctual μXRF analyses were performed for each sample. The limit of detection (LoD) was determined according to the equation [30,31]

$$\text{LoD} = 3.3\ \frac{\sigma}{S} \qquad (1)$$

here σ is the standard deviation of the response and S is the slope of the calibration curve [32]. The standard deviation of the response can be estimated by the standard deviation of either y-residuals or y-intercepts of regression lines. The limit of detection (LOD) for each analysed element (ppm) was: 5 for iron (Fe), 51 for calcium (Ca), 7 for phosphorus (P), 77 for potassium (K), 3 for manganese (Mn), 9 for copper (Cu), 1 for Zinc (Zn), 6 for Sulphur (6), 25 for aluminium (Al), 15 for arsenic (As) and 43 for silicon (Si).

### 2.4. Elements Quantification by ICP-OES

For the quantification of elements in plants, sample preparation was performed by a microwave digestion system (Mars plus CEM, Cologno al Serio Italy). Measurements were done by an ICP-OES (Optima 8000DV, Perkin Elmer Corp., Norwalk, CT, USA) with an axially viewed configuration for element quantification [33], equipped with an ultrasonic nebulizer; the limit of detection (LOD) for each analysed element (ppm) was as follows: 0.006 (Ca, λ 317.9 nm); 0.020 (P, λ 213.6 nm); 0.020 (K, λ 766.4 nm); 0.004 (Fe, λ 259.9 nm); 0.004 (Cu, λ 324.7 nm); 0.006 (Mn, λ 257.6 nm); 0.006 (Zn, λ 213.8 nm); 0.016 (S λ 181.9 nm); 0.004 (Al, λ 308.2 nm); 0.018 (As, λ 193.6 nm); 0.020 (Si, λ 251.6 nm).

About 0.1 mg of powder samples were directly placed into a 100 mL PFA HP-500 Plus digestion vessel (Mars plus CEM, Cologno al Serio, Italy), and 2 mL of 30% (m/m) $H_2O_2$, 0.5 mL of 37% HCl, and 7.5 mL of 69% $HNO_3$ solution were added to the vessel. The heating program was performed in a single step. The temperature linearly increased from 25 to 180 °C in 37 min and was held at 180 °C for 15 min. After the digestion procedure and cooling, the digested samples were diluted to a final volume of 20 mL with Milli-Q water. The analyses were performed in triplicate. The accuracies of the determinations were evaluated by the analysis of tomato leaf Certified Reference Material (CRM 1573a). The technical conditions of ICP–OES analysis are indicated in Table 1.

**Table 1.** Operative conditions of ICP–OES.

| Instrumental Parameters | |
| --- | --- |
| Plasma gas flow | 10 L min$^{-1}$ |
| Auxiliary gas flow | 0.2 L min$^{-1}$ |
| Nebulizer gas flow | 0.55 L min$^{-1}$ |
| RF power | 1450 watts |
| Viewing height | 15 mm |
| Plasma view | Axial |
| Read parameters | Auto |
| Peristaltic pump flow rate | 1.5 mL min$^{-1}$ |
| Processing peak | Height |
| Calibration | Linear calculated intercept |
| Injector Alumina | 2.0 mm i.d. |
| Quartztorch | 1 slot |

### 2.5. µXRF Measurement Strategies

Samples were prepared as pressed powder pellets to allow the retrieval after measurement, producing a homogeneous powder with particles of average size <20µm. The measurements were carried out on a flattened surface of all powders using a live time measurement of 100 s. A fixed amount of powder was placed in cylindrical plastic sample holders with a diameter of 3 mm and a height of 5 mm (Figure 3A,B). The powder was manually pressed to ensure homogeneous packing [26].

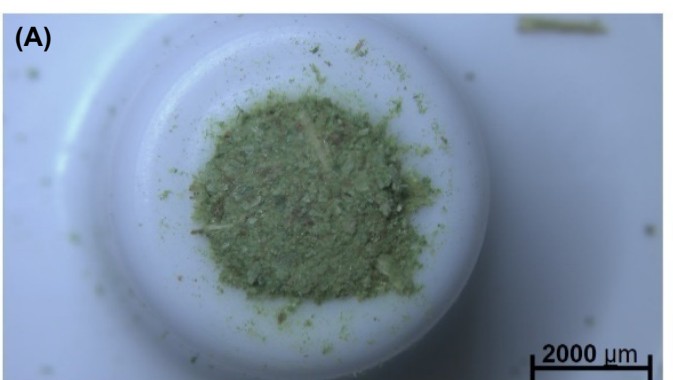
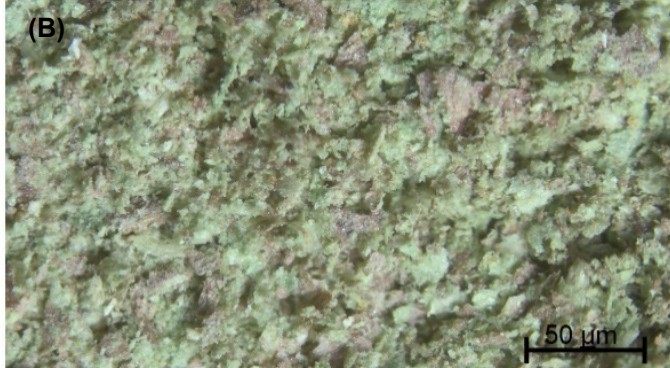

**Figure 3.** Sample preparation into stubs before XRF analysis (**A**). Representative image at the stereomicroscope of powder texture after milling process (**B**).

Examination of the spatial variation in the size of the Rh Compton peak was used to evaluate that packing was homogeneous [26,34]. Means and confidence intervals were calculated based on repeated measurements for each sampling time [35]. To evaluate the effect of sample surface heterogeneity, 30 measurements were executed on the powder surface for each sampling day (samples S1T0, S1T30, S1T60 and samples S2T0, S2T30, S2T45). Subsequently, the µXRF data were compared with the ICP–OES results by principal component analysis.

### 2.6. Principal Component Analysis (PCA)

PCA is a powerful and versatile method capable of providing an overview of complex multivariate data, and it is widely adopted to treat µXRF data [36,37]. PCA can be used for revealing relations between variables and samples (i.e., clustering), detecting outliers, finding and quantifying patterns, and generating new hypotheses. It allows the processed

spectral data to be decomposed into several principal components (PCs), linear combinations of the original spectral data, embedding the spectral variations of each collected spectral data set [38]. According to this approach, a reduced set of factors is produced. Such a set can be used for exploration, since it provides an accurate description of the entire dataset. The first few PCs resulting from PCA are generally used to analyze the common features among samples and their grouping: in fact, samples characterized by similar spectral signatures tend to aggregate in the score plot of the first two or three components. As a multivariate unsupervised statistical procedure, PCA is widely used as an exploratory data tool. In particular, by plotting the principal components (score plots), clusters may appear in the graph, which are indicative of samples with similar composition/spectrum. The loading plot shows the importance of different variables for sample clustering in the score plot. The higher the absolute value of the loading, the more important the variable for the PCA model. In order to better understand the data, it is important to evaluate both score and loading plots at the same time [36,38].

Usually, all the data intended for processing are compiled in matrix form. In this matrix (typically called X), each row contains raw data (variables) describing each sample [39]. In this case study, the variables of XRF data were the concentration values of elements for each sampling day. In geometrical terms, the X matrix represents the data of all studied samples in J-dimensional space, where J is the number of variables describing each sample. PCA is the decomposition of the X matrix into the product of the scores of matrix T and transposed loadings of matrix P plus the residuals of matrix E. The diagram in Figure 4 summarizes how PCA works.

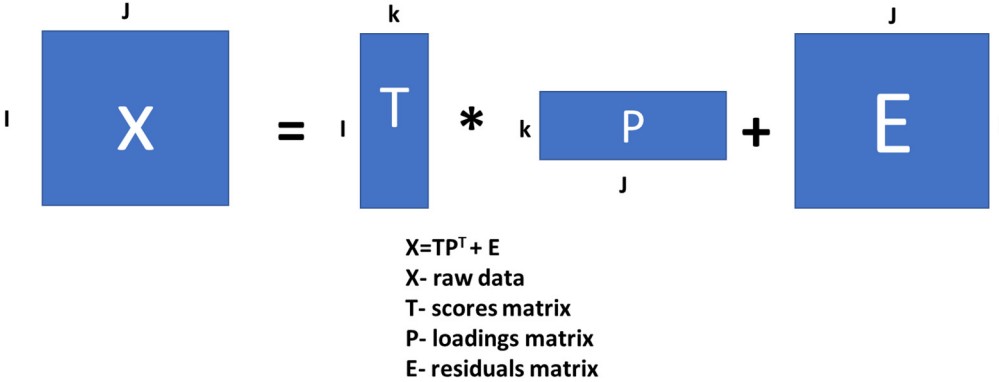

$X = TP^T + E$
X- raw data
T- scores matrix
P- loadings matrix
E- residuals matrix

**Figure 4.** Representation of principal component analysis applied to µXRF data.

These analysed µXRF and ICP data were organized in the 120 × 11 X matrix (120 measurement × 11 elements) for each plant grown in the soils (i.e., S1 and S2). This matrix can be decomposed according to the PCA procedure, and the score and loading matrices can be derived as a result. The size of the scores matrix will be 120 × k, and the size of the loadings matrix will be 8 × k, where k is the number of PCs involved in decomposition [36,40]. These PCs are new geometrical coordinates (axes) that are drawn in the direction of maximum variance (scatter) in the data. Each new PC is orthogonal to all previous PCs.

## 3. Results

### 3.1. Arsenic and Micro- and Macronutrient Concentrations Detected by µXRF and ICP–OES Analyses

For this study, *P. vittata* plants were grown on two natural As-rich soils, characterised by moderate (58 mg/kg DW, S1 soil) and high (750 mg/kg DW, S2 soil) As content and with a significant range of As concentrations in plant tissues (between 2.99 to 3.240 mg/kg DW) to compare the results of µXRF and ICP–OES analyses. The average concentration detected by µXRF technique for As and elements in plants grown in S1 soil for 30 or 60 days is similar to that detected by ICP–OES. Only for some trace elements (i.e., As concentration at time 0) is the detection limit lower for µXRF than for ICP–OES (Table 2).

**Table 2.** Quantitative μXRF and ICP–OES measurement of powder samples obtained from dried fronds of *P. vittata* grown in S1 soil at 0, 30 and 60 days. Results in mg/kg are expressed as the mean with its associated confidence interval.

| T0 | μXRF | ICP-OES | T30 | μXRF | ICP-OES | T60 | μXRF | ICP |
|----|------|---------|-----|------|---------|-----|------|-----|
| Fe | 222.5 ± 32 | 182.9 ± 17 | Fe | 173.0 ± 17 | 174.7 ± 48 | Fe | 122.3 ± 26 | 124.6 ± 5 |
| Ca | 7186.6 ± 620 | 6256.4 ± 863 | Ca | 4700.1 ± 527 | 4440.5 ± 429 | Ca | 3752.6 ± 544 | 3524.4 ± 150 |
| P | 1130.6 ± 139 | 1394.7 ± 265 | P | 1407.9 ± 193 | 1433.6 ± 43 | P | 1482.3 ± 205 | 1505.8 ± 61 |
| K | 4960.0 ± 594 | 5628.4 ± 603 | K | 7173.6 ± 407 | 6980.1 ± 179 | K | 9833.2 ± 728 | 9663.0 ± 2803 |
| Mn | 96.4 ± 12 | 106.3 ± 9 | Mn | 50.9 ± 7 | 53.0 ± 3 | Mn | 31.5 ± 2 | 33.8 ± 1 |
| Cu | 5.5 ± 1 | 24.9 ± 11 | Cu | 88.9 ± 23 | 91.4 ± 5 | Cu | 78.4 ± 16 | 88.6 ± 7 |
| Zn | 49.0 ± 5 | 41,6 ± 3 | Zn | 69.3 ± 4 | 73.2 ± 16 | Zn | 25.4 ± 1 | 27,5 ± 4 |
| S | 4823.4 ± 535 | 5568.6 ± 288 | S | 6715.3 ± 871 | 6727.1 ± 392 | S | 4809.4 ± 635 | 4785.6 ± 55 |
| Al | 45.4 ± 13 | 86.9 ± 21 | Al | 68.7 ± 9 | 76.3 ± 14 | Al | 38.7 ± 8 | 56.6 ± 2 |
| As | - | 3.0 ± 1 | As | 105.7 ± 5 | 108.4 ± 25 | As | 466.9 ± 39 | 468.5 ± 112 |
| Si | 1517.6 ± 551 | 1165.5 ± 231 | Si | 2570.2 ± 661 | 2579.2 ± 400 | Si | 1541.5 ± 204 | 1532.5 ± 491 |

Similarly, comparable results were obtained from ICP and μXRF measurements of As and other elements in *P. vittata* tissues of plants grown in the S2 soil (Table 3).

**Table 3.** Quantitative μXRF and ICP–OES measurements of powder samples obtained from dried fronds of *P. vittata* grown in S2 soil for 0, 30 and 45 days. Results in mg/kg are expressed as the mean with its associated confidence interval.

| T0 | μXRF | ICP-OES | T30 | μXRF | ICP-OES | T45 | μXRF | ICP |
|----|------|---------|-----|------|---------|-----|------|-----|
| Fe | 175.1 ± 42 | 186.3 ± 13 | Fe | 142.5 ± 29 | 144.2 ± 18 | Fe | 94.3 ± 23 | 96.6 ± 8 |
| Ca | 6222.8 ± 705 | 5095.5 ± 897 | Ca | 5521.1 ± 543 | 5221.6 ± 110 | Ca | 5138.3 ± 518 | 4896.8 ± 62 |
| P | 1370.8 ± 122 | 1184.4 ± 222 | P | 1077.2 ± 115 | 1101.7 ± 28 | P | 909.6 ± 125 | 951.9 ± 42 |
| K | 4774.0 ± 278 | 5228.7 ± 441 | K | 7726.4 ± 798 | 7524.7 ± 427 | K | 6454.6 ± 554 | 6328.7 ± 158 |
| Mn | 100.8 ± 18 | 130.8 ± 18 | Mn | 78.8 ± 13 | 80.3 ± 10 | Mn | 51.6 ± 10 | 53.6 ± 1 |
| Cu | 16.5 ± 3 | 20.5 ± 3 | Cu | 19.4 ± 3 | 25.0 ± 2 | Cu | 14.7 ± 3 | 24.4 ± 1 |
| Zn | 38.8 ± 4 | 41.7 ± 2 | Zn | 44.0 ± 2 | 46.1 ± 2 | Zn | 49.4 ± 5 | 51.9 ± 3 |
| S | 5481.6 ± 394 | 5417.9 ± 234 | S | 7350.0 ± 795 | 7250.5 ± 229 | S | 5303.7 ± 693 | 5311.6 ± 111 |
| Al | 51.9 ± 9 | 83.7 ± 18 | Al | 49.0 ± 10 | 59.0 ± 20 | Al | 14.7 ± 4 | 27.1 ± 5 |
| As | - | 5.8 ± 2 | As | 17169 ± 139 | 1651.7 ± 125 | As | 3357.2 ± 302 | 3240.5 ± 245 |
| Si | 1153.4 ± 224 | 945.4 ± 225 | Si | 2745.2 ± 816 | 2725.2 ± 811 | Si | 1935.1 ± 367 | 1940.5 ± 257 |

The similarity of the data obtained by the standard quantitative technique ICP–OES and by μXRF demonstrate the reliability and accuracy of μXRF. We also performed PCA on the data obtained by ICP–OES and μXRF to assess nutrient content in relation to As accumulation.

In this study, we have analysed powdered samples that were not subject to any treatment before analysis. μXRF technology has recently been used to measure micro- and macroelements as well as fertilisers in plants of agronomic interest, achieving good precision and low detection limits [41]. However, to date, there are no studies performed on As hyperaccumulators to monitor As in plant tissues in which there is comparison between the results of μXRF and ICP–OES.

### 3.2. PCA Models of T0 Samples for μXRF and ICP–OES Analysis

The slight difference in the average amount of each element in T0 samples could be due to *the* intrinsic variability of selected plants. To demonstrate that this variability is not significant, PCA of μXRF and ICP–OES was performed (Figure 5). In detail, T0 S1 and T0 S2 samples show overlap and are within the confidence level produced by the PCA model. The loading plots show which elements define the distribution of the data displayed in the PCA score plot. These results demonstrate the representativeness of the selected plants

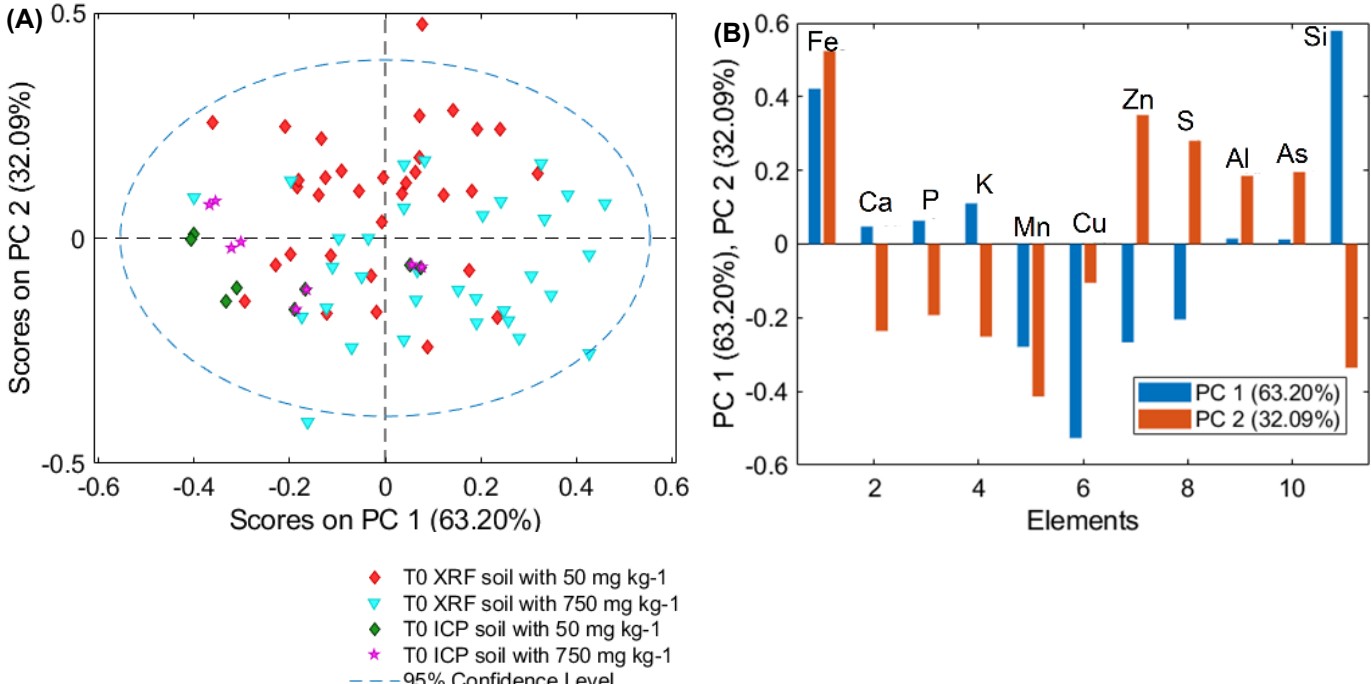

**Figure 5.** PCA (**A**) score and (**B**) loading (PC1, PC2) plots of µXRF and ICP–OES data for T0 samples for plants grown on soil S1, S2.

### 3.3. PCA Models of µXRF and ICP–OES Analysis

PCA models were created for evaluating the variance of each µXRF with ICP–OES measurements. The PCA score plot of fronds grown on S1 soil shows a good level of overlap between the ICP–OES data and the µXRF data obtained for each sample. In detail, PCA, applied to the pinna powder data, shows good separation for each time point (Figure 6). The total variance captured was 99.91%, with five principal components (PC). The score plot of PC1, PC3 and PC5 highlights three different clouds corresponding to three different compositions of the examined powders. PC1 allows evaluating the differences between the element content of pinna powders at times 0 and at times 30 and 60. PC3 highlights the difference between pinna powders at times 30 and other sample times. Lastly, PC5 shows the greatest variation for an increase in As (Figure 6A). The loading plot allows for interpretation of the observed grouping in terms of elemental composition. In detail, by analysing the loading plot, it is possible to highlight the weight of the macro and micronutrients detected in the pinna powders for each sampling day considered. The positive values of PC1 are mainly due to K, while the negative values are mainly due to calcium (Ca) and sulphur (S). The positive values of PC3 are mainly due to Si, while the negative values of PC3 are due to S, potassium (K), phosphorus (P), and Ca. Lastly, the positive values of PC5 are mainly due to the variance in As, as well as in Cu and Fe, while negative values are mainly due to the combined variance of K, P, Ca, S and silicon (Si) (Figure 6B).

The PCA score plot of samples grown in the S2 soil shows a good level of overlap between the ICP–OES data and the µXRF data obtained for each sample.

Similar to the results obtained with soil S1, PCA applied to pinna powder data shows good separation for each time point for both ICP–OES and µXRF data (Figure 7). The total variance captured was 99.78% with 5 PCs. The score plot of PC1, PC2 and PC3 highlights three different clouds corresponding to the three different compositions of the examined powders (Figure 7A).

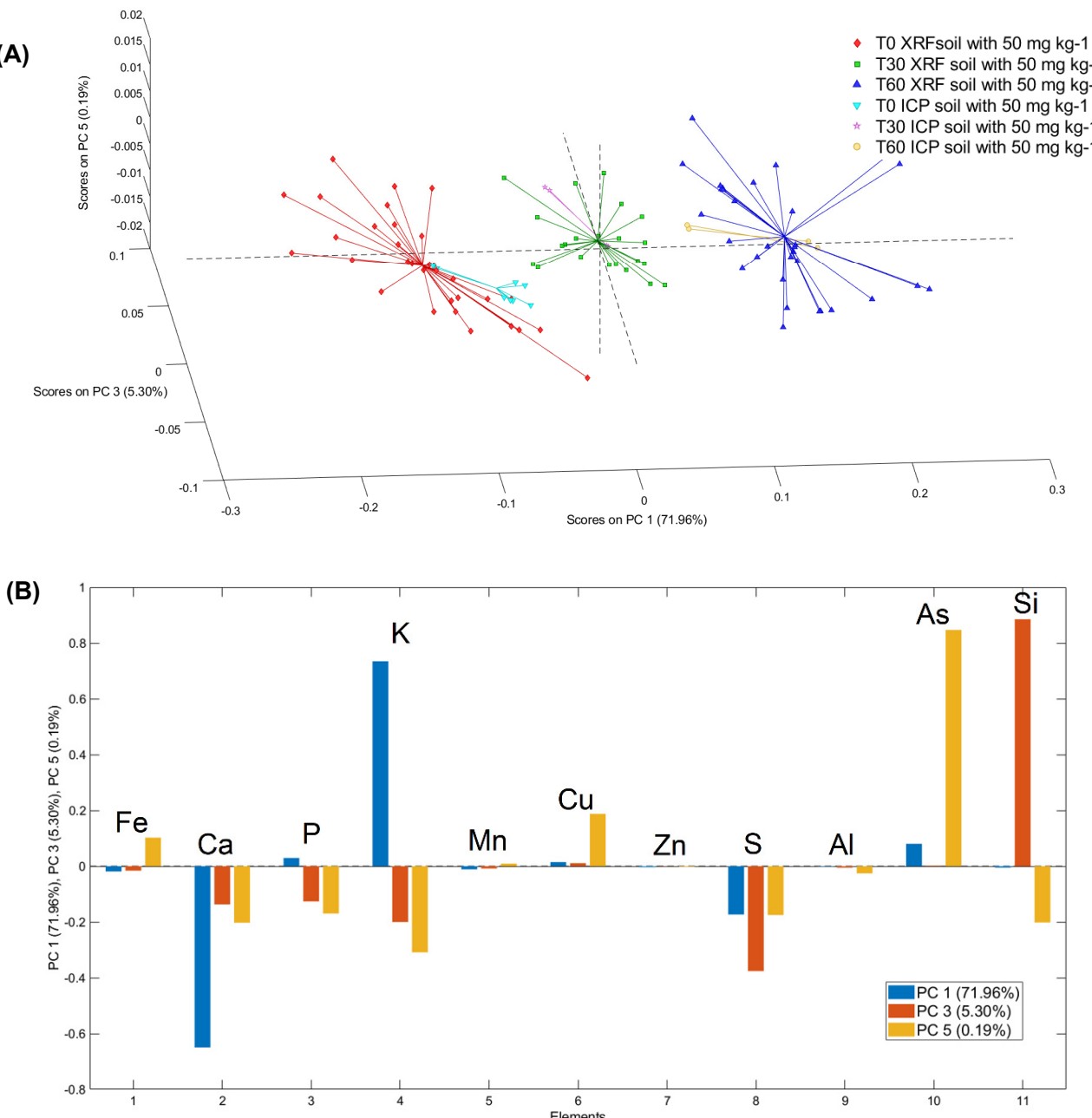

**Figure 6.** 3D PCA (**A**) score and (**B**) loading (PC1, PC3, PC5) plots of μXRF and ICP–OES data for T0, T30 and T60 for plants grown on soil S1.

The PC1 allows for evaluation of the differences in the content of the elements detected in the samples at time 0 in comparison with those at 30 days and 45 days. The second and third PCs highlight the differences in the sample at 30 days compared to all the other time points.

The loading plot allowed for interpretation of the observed grouping in terms of elemental composition. In detail, by analysing the loading plot, it is possible to highlight the weight of the macro- and micronutrients detected in the pinna powders for each of the considered sampling days. In detail, the positive values of PC1 are mainly due to As, K and Si, while the negative values are mainly due to Ca, S, and P. The positive values of PC2 are mainly due to S and Si, while the negative values of PC2 are due to Ca, K and P. Lastly, the positive values of PC3 are mainly due to the variance in K, S and P, while negative values are mainly due to the variance in Ca, Si and As (Figure 7B).

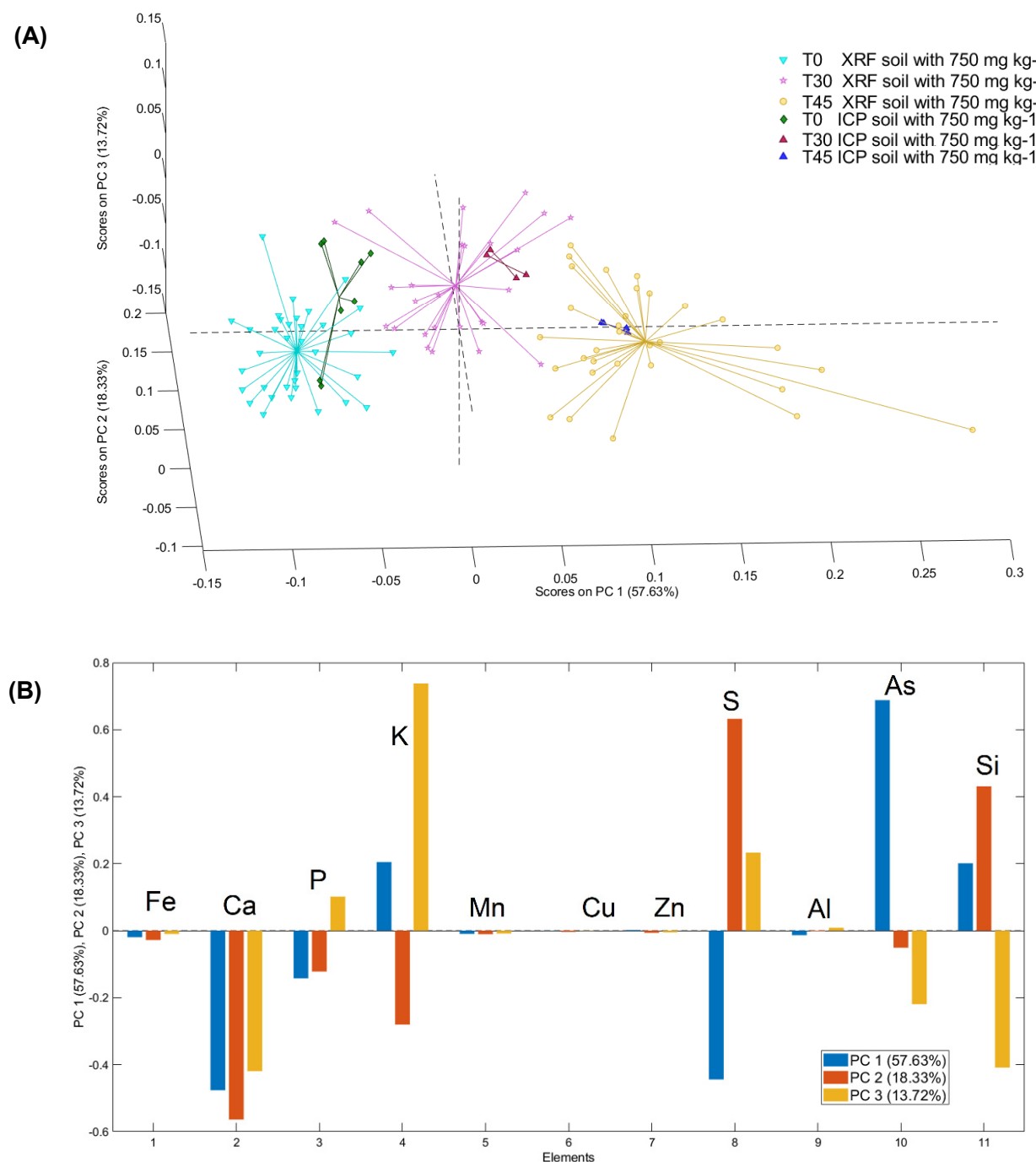

**Figure 7.** 3D PCA (**A**) score and (**B**) loading (PC1, PC2, PC3) plots of the μXRF and ICP–OES data for T0, T30 and T45 for plants grown on soil S2.

Different studies have demonstrated the reliability of XRF to quantitatively measure certain elements in plant tissues, such as Pb, Cr, and Ni, with a good correlation between XRF and ICP data [14,42,43]. Furthermore, several authors have demonstrated that bench-top μ-XRF is a versatile tool for plant analysis. In fact, it can be used to determine the elemental mineral composition of a limited area of plant tissue (measuring approximately 30 μm) produce lines and 2D images [18,44,45]. This allows the monitoring of nutrient uptake kinetics [44].

Toxic elements, especially As and its compounds, may strongly influence the metabolic processes of plants [46]. However, we have shown that by combining XRF, FP and PCA, it is possible to perform quantitative estimation of the elements with *a* correlation between As

content and nutrient concentrations. This information is useful for monitoring the health of plants during As accumulation and assessing possible alterations in the phytoextraction process resulting from nutritional deficiencies in the plant.

In detail, the μXRF and ICP–OES measurements, validated by PCA analysis, show that in *P. vittata* fronds collected from plants grown on both S1 and S2 soils, the K content increased over time in relation to the As increase, whereas the P content increased at moderate As concentrations and decreased at high As concentrations. On the other hand, Fe, Ca and Mn contents decreased over time in relation to the As increase. In addition, a rapid increase in Si and S content was observed at moderate As, but their levels returned to the initial levels at high As concentration (Tables 2 and 3). These data are in agreement with those of Tu et al. [47], where As content and element variation in P. vittata were evaluated using conventional techniques such as GFAAS and ICP–AES.

It is well known that P and K macronutrients are critical for plant metabolism. Potassium is also related to active protein synthesis and has an osmotic role. In non-hyperaccumulator plants such as tomatoes [48,49], As reduces the uptake of both macronutrients such as K as well as micronutrients such as boron (B), Cu, Mn and Zn. In contrast, in *P. vittata*, the As-related increase in K is probably a response and attempt to balance the excess of anions caused by As hyperaccumulation. A similar suggestion has been proposed by Tu et al. [47]. Our unpublished data from μXRF-derived mapping show a similar distribution of As and K, in agreement with Lombi et al. [50], thus providing further evidence for its role as a countercation for As in plants.

It is well known that P in plants is important for energy transfer and protein metabolism. Arsenate, which is a phosphate (Pi) analogue, is uptaken via the Pi transporters in higher plants, including in *P. vittata*, and thus competes with P in the plant uptake phase [51–53]. Tu et al. [47] have shown that the addition of As to *P. vittata* hydroponic culture increases P content, especially in young fronds. By evaluating the As response in fronds of P. vittata grown in soil naturally rich in As in our study, we observed a P increase in S1 samples, in which the As concentration spanned from 100 to 470 mg/kg DW (Table 2). In agreement, Tu et al. [47] found an increase in P in the fronds of *P. vittata* plants hydroponically grown at similar As concentrations. This may be due to competition between As and P in the soil, as competitive As adsorption causes the release of bioavailable P [54,55], as well as in the plant, where As induces an increase in P requirement [48,56].

In contrast, the decrease in P accumulation observed in fronds collected from P. vittata plants grown in S2 soil, where the As concentration spans from 1700 to 3300 mg/kg DW (Table 3), is probably due to competitive As uptake [51].

Calcium is a macronutrient essential for membrane permeability and cell integrity [57]. As accumulation negatively influenced Ca accumulation in *P. vittata* fronds, mainly at high As levels (Tables 2 and 3), suggesting that Ca has a limited role in the response of *P. vittata* to As toxicity.

Iron and Mn are micronutrients mainly required as constituents of prosthetic groups in metalloproteins and activators of enzyme reactions.

The arsenic-related reduction in micronutrients observed in the fronds is probably a result of As phytotoxicity.

In agreement with previous data obtained in barley and rice, an As-related reduction in Fe was observed in *P. vittata* fronds. A possible explanation of this is that As could decrease Fe translocation from the roots to the shoots by competing with proteins involved in Fe translocation [58].

The observed As-related S increase in both S1 and S2 samples can be partly due to increased contents of glutathione (GSH), which is involved in antioxidant responses countering oxidative stress, as shown in Wei et al. [59]. In any case, no direct S increase with either moderate or high As was observed. It was also reported that As can enhance GSH biosynthesis in *P. vittata* fronds without modulating the S concentration [59].

Silicon content does not appear to be closely correlated with As increase. While Ma et al. [60] showed that As$^{III}$, as a Si analogue, uses the same silicon efflux transporter,

Low silicon 1, Lsi1, for entry into rice roots and the same silicon efflux transporter, Low silicon 2, Lsi2, for translocation to the shoot via xylem loading, different channels are utilised in *P. vittata* for the uptake of As[III] and entry into the roots. In addition, our results are in good agreement with previous data showing that As[III] does not influence Si uptake in *P. vittata* grown in hydroponics [61].

Data obtained from S2 samples showed the same elemental variation observed in S1 samples, thus confirming that Fe, Ca, Mn, P and K variations are strictly related to As concentrations in plant fronds. To rule out the possibility that the observed element variation is related to soil characteristics, *P. vittata* S1 and S2 samples containing similar As concentrations (288.29 ± 45 and 319.69 ± 21, mg/kg DW) were compared for the evaluation of Fe, Ca, Mn, P, S and K contents (Table 4). As expected, the Fe, Ca, P and Mn contents do not show significant differences between S1 and S2 samples, thus indicating that variations in these elements are mainly related to As content. In contrast, the average K and S contents showed significant differences, possibly because of their involvement in critical processes for plant metabolism.

**Table 4.** Macro- and micronutrient contents (mg/kg ± Standard Error (SE)) measured by μXRF, in samples collected from *P. vittata* plants grown on S1 and S2 soils, characterised by similar As content.

| Macro- and Micronutrient | Quantity in S1 Samples | Quantity in S2 Samples |
|---|---|---|
| As | 319.7 ± 22 | 288.4 ± 45 |
| Fe | 227.5 ± 40 | 228.4 ± 49 |
| Ca | 4296.6 ± 289 | 5369.2 ± 554 |
| P | 1047.9 ± 125 | 1001.9 ± 113 |
| K | 8595.6 ± 619 | 6413.7 ± 485 |
| Mn | 100.86 ± 14 | 104.5 ± 19 |
| S | 3975.13 ± 421 | 8491.7 ± 434 |
| Cu | 93.8 ± 26 | 12.7 ± 2 |

Regarding Cu content, a rapid increase at moderate As concentration (about 100 mg/kg DW, Table 2) was detected in S1 samples, but there was no further change at high As content (spanning from 1700 to 3300 mg/kg DW, Table 3) in S2 samples. These data confirm that Cu content is not directly related to As concentration. Thus, the limited increase in Cu content could be due to changes in plant metabolism in response to the oxidative stress induced by As, as proposed by Farnese et al. [62].

## 4. Conclusions

In this study, data obtained by μXRF show a high correlation with ICP–OES data, also confirming the reliability of this green analytical method (GAM) in detecting low As concentrations in a quantitative way, with the exception of trace amounts (i.e., As concentrations at time 0). Furthermore, as μXRF is a non-destructive technology, it allows the sample to be used for further analysis with other analytical methods (Figure 8). The principal component analysis (PCA) score plots from the ICP–OES and μXRF measurements directly highlight the response of the plant to As contamination. In fact, the cloud variation of each time point is not only dependent on the As increase but also on the variation of the element concentration with increasing As. In addition, PCA confirms the good fit between ICP–OES and μXRF results.

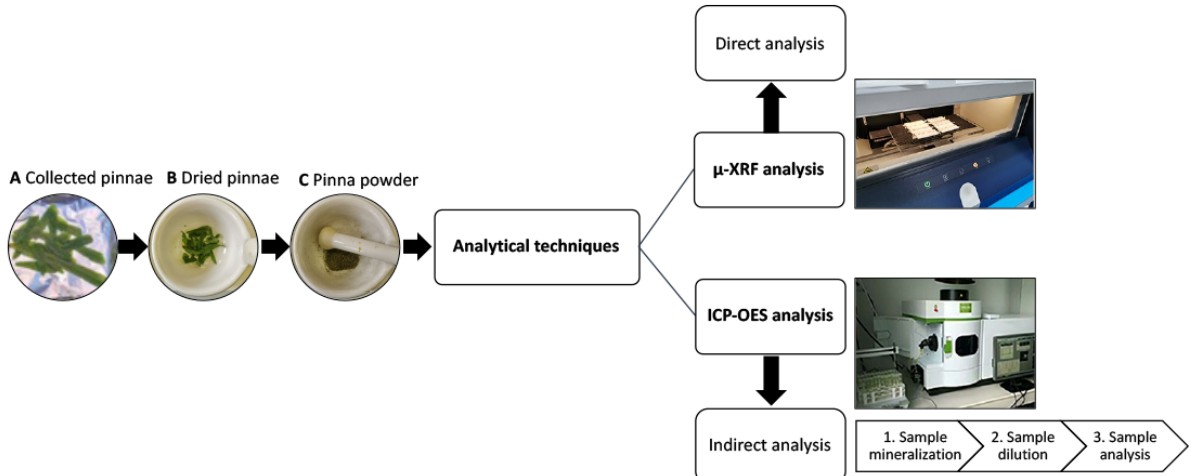

**Figure 8.** Flowchart demonstrating technical steps used in the approaches of μXRF and ICP–OES analyses.

The proposed methodological approach (i.e., the combination of pinna powder, FP and PCA) ensures the high representativeness of the samples, and for this reason, it can be utilised on portable devices allowing accurate in situ analysis.

In addition, the μXRF and ICP–OES measurements showed that changes in the contents of most nutrients in plant tissues are related to As amount rather than As soil concentration, with the exception of S and K, whose content is mainly related to the plant metabolism.

Furthermore, this study proposes a μXRF-based analytical method for quantitative As and macro- and micronutrient determination in *P. vittata* plants, combined with a chemometric approach. Moreover, PCA was performed to validate the correlation between the As content and the macro- and micronutrient contents in *P. vittata* fronds. The possibility of using portable instruments based on this methodological approach will allow its application in monitoring phytoextraction processes in the field. The proposed methodological approach applied to the hyperaccumulator *P. vittata* allows the fern to be used for the biomonitoring of contaminated soils or for phytoextraction processes. Other methods used for measuring elements in plants, though more sensitive and accurate technologies, are very often slow, laborious and expensive. In conclusion, μXRF-based techniques allow rapidly measuring element concentrations in plants at the lab scale and obtaining quantitative measurements in near-real-time. This approach is also promising for large-scale measurements of plants grown in the field.

**Author Contributions:** Conceptualization, investigation S.S., G.B., G.C. and P.B.; methodology, G.C., R.M. and M.L.A.; validation, G.C., S.R.S. and P.B.; formal analysis, G.C., R.M. and M.L.A.; writing—G.C. and P.B.; supervision, writing—review and editing G.B., S.S. and M.C. All authors have read and agreed to the published version of the manuscript.

**Funding:** This research was funded by Regione Lazio, 85-2017 FONDI LISPA 15067 CUP. B56C180008 70002.

**Institutional Review Board Statement:** Not applicable.

**Informed Consent Statement:** Not applicable.

**Data Availability Statement:** Not applicable.

**Conflicts of Interest:** The authors declare no conflict of interest.

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
