# Peer review of "A Green Approach Based on Micro-X-ray Fluorescence for Arsenic, Micro- and Macronutrients Detection in Pteris vittata"

_water, doi:10.3390/w14142202_

Round 1

Reviewer 1 Report

This paper took an excellent trial to detect As and other elements in Pteris vittata using benchtop µXRF. However, this manuscript needs further improvement.

1.     AsV and AsIII are not accurate in showing arsenate and arsenite. As(V), As(III), or Asv, AsIII is often used.

2.     µXRF is usually used to detect the composition in situ. When it is used to measure powder samples, how did you ensure the homogeneity of the sample? What’s the size of your fine particle samples? Since M4 Tornado possesses spot size <20 µm, if the powder is >20 µm, the results should be pretty inhomogenous.

3.     Please check the kg/m3 in line 127.

4.     Please use the standard expression for a decimal point, not a comma.

5.     It is necessary to compare the µXRF and the ICP results, not just show the original data. For example, the Fe concentration for T0 in Table 1 is 222.5 and 182.85 mg/kg obtained from µXRF and ICP, respectively, and the first data is 121% of the latter. To do so, the readers can quickly know the accuracy of µXRF data.

6.     It is necessary to keep the same significant digit for the same element obtained using the same method.

7.     The method of the PCA model should be described in the method section.

8.     The style of the unit should be kept uniform. Currently, both mg/kg and mg kg-1 are used.

9.     In my opinion, the two T0 should be the same. But they have different element concentrations and PCA model, why?

10.   There is no Cu data in Table 3. The Fe, Ca, P, and Mn could not be in the same group, since Ca in the S2 is 25% higher than in S1 while Fe, P, and Mn are almost identical.

11.   The equipment used in reference 11 is field portable XRF, not µXRF. Please check other references as well. Otherwise, it’s another story.

Author Response

Indeed, the reviewer’s comments and suggestions helped us in substantially improving the quality of our manuscript. We hope these changes and improvements are satisfactory.

English language and style have been improved.

Introduction, References, Methods, Results and Conclusions have been improved.

References to the manuscript refer to the Revised Manuscript

This paper took an excellent trial to detect As and other elements in Pteris vittata using benchtop µXRF. However, this manuscript needs further improvement.

  1. AsV and AsIII are not accurate in showing arsenate and arsenite. As(V), As(III), or Asv, AsIIIis often used.

Corrections were made according to reviewer suggestions

  1. µXRF is usually used to detect the composition in situ. When it is used to measure powder samples, how did you ensure the homogeneity of the sample? What’s the size of your fine particle samples? Since M4 Tornado possesses spot size <20 µm, if the powder is >20 µm, the results should be pretty inhomogenous.

Thank you for the suggestion. We have implemented the Methods section by describing how we have obtained a homogeneous powder with average particle size <20 µm (Par 2.5 Figure 3 of the Revised Manuscript). Anyway, according to most references the M4 Tornado spot size is  ~30 µm (Rodriguez et al., 2011)

  1. Please check the kg/m3in line 127.

Thank you, correction was made

  1. Please use the standard expression for a decimal point, not a comma.

Corrections were made according to reviewer suggestion

  1. It is necessary to compare the µXRF and the ICP results, not just show the original data. For example, the Fe concentration for T0 in Table 1 is 222.5 and 182.85 mg/kg obtained from µXRF and ICP, respectively, and the first data is 121% of the latter. To do so, the readers can quickly know the accuracy of µXRF data.

Thank you for the suggestion. Plants have a high intrinsic variability and for this reason in addition to the analysis of average values and confidence intervals, the Principal Components Analysis (PCA) was performed. PCA is used to compare all data acquired by ICP and XRF, and to assess their variance and accuracy.

The description of the PCA model was implemented in the Methods Section (par 2.6, of the Revised Manuscript)

  1. It is necessary to keep the same significant digit for the same element obtained using the same Corrections were made according to reviewer suggestions
  2. The method of the PCA model should be described in the method section.

The description of the PCA model was implemented in the Methods Section (par 2.6 of the Revised Manuscript)

  1. The style of the unit should be kept uniform. Currently, both mg/kg and mg kg-1are used.

Corrections were made according to reviewer suggestion

  1. In my opinion, the two T0 should be the same. But they have different element concentrations and PCA model, why?

We agree with the reviewer that the values T0 should be the same. However, the slight differences observed are probably due to the intrinsic variability between the selected plants. To demonstrate that there are not significant differences between the values, we performed a PCA referring to µXRF and ICP data for T0 S1 and S2 samples (Results Section par 3.2 Figure 5 of the Revised Manuscript))

  1. There is no Cu data in Table 3. The Fe, Ca, P, and Mn could not be in the same group, since Ca in the S2 is 25% higher than in S1 while Fe, P, and Mn are almost identical.

We thank the reviewer for this comment. In the new version Cu was included in the table. As for Ca,  the confidential limit of the two average amounts is  4008,7- 4584,4; 4814,8- 5923,5. Thus there is only a slight significant difference possibly due to the different physiological state of the plants such as the different age of the fronds. Infact it was shown that the highest Ca concentrations are usually found in old leaf tissues, which show a higher rate of transpiration responsible for Ca translocation from the roots to these organs. (Tu et al., 2005)

  1. The equipment used in reference 11 is field portable XRF, not µXRF. Please check other references as well. Otherwise, it’s another story.

The text was modified and the reference 11 was changed as suggested by the reviewer.

Reviewer 2 Report

This study adopts benchtop micro-X-ray fluorescence spectrometry (µXRF) method ro determine the content of As, micro- and macro-nutrient elements concentrations in P. vittata, with comparison to determination results of ICP-OES for confirming reliability of µXRF. PCA was used to test the correlation between As and other elements concentrations. However, as authors stated, µXRF and ICP-OES mehod has been adopted to measure heavy metals contents in plant tissues, so in this study only using P. vittata frond as test plant appears to be low innovations. Also, this manuscript is in poor presentation, especially some sentences are hardly understood and/or no sense, the language should be checked around whole manuscript. How to run PCA analysis is not decribed in Methods section, but correlation of As and other elements content in P. vittata had been reported previously, and this study does not include discussion with novel insights.

Author Response

REVIEWER 2

Indeed, the reviewer’s comments and suggestions helped us in substantially improving the quality of our manuscript. We hope these changes and improvements are satisfactory.

English language and style have been improved.

Introduction, References, Methods, Results and Conclusions have been improved.

References to the manuscript refer to the Revised Manuscript

This study adopts benchtop micro-X-ray fluorescence spectrometry (µXRF) method ro determine the content of As, micro- and macro-nutrient elements concentrations in P. vittata, with comparison to determination results of ICP-OES for confirming reliability of µXRF. PCA was used to test the correlation between As and other elements concentrations.

1) However, as authors stated, µXRF and ICP-OES methods has been adopted to measure heavy metals contents in plant tissues, so in this study only using P. vittata frond as test plant appears to be low innovations. Also, this manuscript is in poor presentation, especially some sentences are hardly understood and/or no sense, the language should be checked around whole manuscript. 2)How to run PCA analysis is not decribed in Methods section, 3) but correlation of As and other elements content in P. vittata had been reported previously, and this study does not include discussion with novel insights.

1) We agree with the Reviewers critical issue. However, the novelty of our work, compared to previous studies, is the methodological approach, which consists of i) the production of fine plant homogeneous powders; ii) the use of µXRF, which in contrast to ICP is a non-destructive technique, and which allowed the analysis to be repeated on the same samples by ICP-OES. Furthermore,  sample preparation for µXRF analysis is simplified because no chemical treatment is required, as indicated in the introduction, in the Methods, and Results (Lines 56-59, Lines 116-117,  275-278)

2) The description of the PCA model was implemented in the Methods Section (par 2.6, Figure 4)

3) In addition, the data obtained were further analysed to correlate the nutrients content with the concentration of As. This was done by Tu et al. 2005,  who, however, analysed young and old fronds and by means of  ICP only, while we examined the average value in the fronds by both ICP-OES and µXRF.

Reviewer 3 Report

Dear authors

Please, find in attached file my comments and suggestions about your manuscript. Due to the lack of information and misunderstandings, I suggest modifications before considering your manuscript for publication in the current journal.

Author Response

REVIEWER 3

Indeed, the reviewer’s comments and suggestions helped us in substantially improving the quality of our manuscript. We hope these changes and improvements are satisfactory.

English language and style have been improved.

Introduction, References, Methods, Results and Conclusions have been improved.

If not specified the number of lines are referred to the previous version

Line 61-62 A confusion was made by the authors in this sentence. Please, correct according to the real information in reference 11. 

Line 63 Why the authors used ICP as reference. They should compare µXRF to various methods like GFAAS, HGAAS, ICPMS......

The aim of this work is to compare semi-quantitative values obtained by µXRF with those obtained by a standard technique for quantitative measurements, such as ICP-OES. This technique was chosen because it has previously been used successfully on plant tissues (Pendergrass et al. 2006; Baldwin et al. 2007; Antenozio et al. 2021; Saba et al. 2020; Janchevska et al. 2020; Kiani A. et al. 2021; Ahmad et al. 2021).

Line 64 The authors should point out the limitation in using XRF when soil moisture is too high.

We agree with the referee.  However, in this work we did not analyse soils but only dried plant tissues. However to follow reviewer remark we added a statement in Introduction of the Revised Manuscript (lines 93-95).

Line 71 Determination of As using µXRF is not a new method. What is the novelty? Please, explain for a better undertanding.

The novelty of our work, compared to previous studies, is the methodological approach, which consists of i) the production of fine plant homogeneous powders; ii) the use of µXRF, which in contrast to ICP is a non-destructive technique, and which allowed the analysis to be repeated on the same samples by ICP-OES. Furthermore, sample preparation for µXRF analysis is simplified because no chemical treatment is required.

In addition, the data obtained were further analysed to correlate the nutrient content with the concentration of As. This was done by Tu et al. 2005, who, however, analysed young and old fronds and by means of  ICP only, while we examined the average value in the fronds by both ICP-OES and µXRF.

Line 75 The attractiveness of this analysis for heavy metals evaluation in plant is mainly related to the possibility to evaluate macro and micro nutrients changes in response to heavy metals content and monitoring overtime their phytoextraction process in different plant tissues.

This is not possible using an other analytical technique?

Heavy metals evaluation in plant is possible by means of analytical techniques such as such as atomic fluorescence spectroscopy (AFS), graphite furnace atomic absorption (GFAA), hydride generation atomic absorption spectroscopy (HGAAS), inductively coupled plasma-atomic emission spectrometry (ICP-AES), Inductively Coupled Plasma-Optical Emission Spectroscopy (ICP-OES) and inductively coupled plasma-mass spectrometry (ICP-MS), while monitoring heavy metal concentration in a plant tissue overtime required non disruptive technics and to date XRF allows a fast and non-destructive analysis, and repeatable measurements on the same samples.  In addition this provides an analysis on a more representative quantity of sample compared to the stated classical analytical approaches.

Line90 This methodology can be applied using As concentrations from ICP-OES.

Some questions remain at this stage :

1) what is the main objective of this study

2) Was µXRF use for the determination of macro and micronutrient?

3)Why the authors did not make a comparison from two correlations, one between macro/micronutrient and As from ICP-OES and the second using As from µXRF?

1) The aim of our study is to correlate As and nutrients accumulation in fronds of Pteris vittata plants grown on two different naturally As-rich soil, by means of a green and fast µXRF-based technique in comparison with the standard quantitative ICP-OES technique. This is clearly stated in the introduction (lines 95-105).

2) µXRF technology can be used for macro and micro nutrients detection (Turner et al., 2018;  Mijovilovich et al, 2020, Feng et al. 2021 Critical Review). Furthermore,  it can be used also for in field application.

3) Indeed, this was performed by means of PCA analysis, as clearly stated in the Introduction:  “Principal Component Analysis (PCA) was used to evaluate the correlation between the content of As and macro and micronutrient in P. vittata fronds”, (lines 102-104 of the Revised Manuscript),’ as well as in the Results: “µXRF and ICP-OES measurements validated by PCA analysis, showed that in P. vittata fronds -collected from plants grown on both S1 and S2 soils….” (lines 342-350 of the Revised Manuscript)

In order to improve the PCA analysis data we have added more technical information about the statistical approach in par 2.6 of Methods of the Revised Manuscript.

METHODS

Line 101 Why the third time step is not the same?

This is because we wanted to compare samples with an As concentration within a certain range (100-3000 mg/Kg). To achieve this from different soils, we collected  samples at different times and selected those of interest, as indicated in the Results (par3.1 lines 245-251 of the Revised Manuscript)

Line118 How was these dried leaves powder prepared?

This has been clarified in the new version as the statement 'performed on dried fronds that have been ground into a fine powder' has been changed to: ' A pinna for each frond was collected from 4 different plants, dried for 48h at 37°C and then ground in a mortar into a fine powder. ‘ (line 116-117 of the Rivised Manuscipt)

Title of 2.1 par has been changed to “Plant growth and pinna powder preparation”

Title of 2.3 par has been changed to 'μXRF analysis on dried fronds powders'

Line 119 For a better understanding, the authors should give detail about ICP-OES and µXRF in the same part.

We agree with the reviewer, the ICP-OES and µXRF methods have been described in more details in par 2.4 -2.5-2.6 of Methods

Line126 This unit is very strange. Unit of concentration is mg/kg

Unit changed to mg/kg

Line 140 Which one? What was the certified sample used and the results obtained using ICP-OES and µXRF?

As regards the ICP-OES, tomato leaf Certified Reference Material (CRM 1573a) was used as reference, as indicated in lines 173-174.   As regards µXRF, Spectrum energy calibration was performed, by using Bruker® calibration standard as indicated in lines 176-178 of Revised version. The quantification was performed by using fundamental parameters as indicated in lines 139-141 of the Revised Manuscript.

What about the water content in plants?:

All the analysis has been performed with dried samples. Anyway, the water content of Pteris vittata fresh frond tissues is in the range of 80-90 % of the total fresh weight, according to the hydration condition of the plants.

Line 154 The authors should indicate the number of replicates (legenda Tabelle 1, 2)

All the ICP-OES analysis has been performed in triplicate as indicated in Methods Line 174 of the Revised Manuscript.

While for the µXRF analysis 30 measurements for each sampling day have been performed, as indicated in lines 195-199 of the Revised Manuscript.

Tabella 1, 2 (Table 2, 3 of the revised version):

The authors have to apply statistical tests in order to affirm if the metal concentrations obtained usinf both analytical methods are staistically differents or not.

For the statistical analysis we have performed Principal Components Analysis (PCA). Plants have a high intrinsic variability and for this reason in addition to the analysis of average values and confidence intervals, the PCA was performed. PCA is used to compare all data acquired by ICP and XRF, and to assess their variance and accuracy.

The description of the PCA model was implemented in the Methods Section (par 2.4, 2.6)

468,4906±112

How do the authors explain this very high SD?

The High SD value is due to the lower number of replicates performed for ICP analysis compared to XRF analysis.  However, PCA analysis shows that all the ICP-OES values are inside the confidence interval defined by μXRF data (Figure 5- 6-7)

The authors have to add information about RSD for each element and should discuss about obtained values.

We have chosen to indicate the Confidential Interval of each μXRF and ICP-OES measurements, as indicated in Table 2 and Table 3. Then we have performed PCA analysis to evaluate the variance of all data in one shot.  

As mentioned above the description of the PCA model was implemented in the Methods Section (par 2.4, 2.6)

Line 164 What is this good correlation in the current study? The authors have to use statistical tests and indicators to demonstrate

As mentioned above this was performed by means of PCA analysis, as clearly stated in the Introduction:  “Principal Component Analysis (PCA) was used to evaluate the correlation between the content of As and macro and micronutrient in P. vittata fronds”, (lines 102-104 of the Revised Manuscript),’ as well as in the Results: “µXRF and ICP-OES measurements validated by PCA analysis, showed that in P. vittata fronds -collected from plants grown on both S1 and S2 soils….” (lines 342-350 of Revised Manuscript)

In order to improve the PCA analysis data we have added more technical information about the statistical approach in par 2.6 of Methods.

Line 164 ….These results demonstrate the reliability of…. (Which one?)

Line165 …..Indeed, Gallardo et al. [24] proposed  ….Why indeed?? The authors have to demonstrate this reliability before comparing their results with the literrature.

We agree with the reviewer that this conclusion must be improved. Thus, the statements have been changed and moved as follows:

“The similarity of the data obtained by the standard quantitative technique ICP-OES and by µXRF demonstrate the reliability and accuracy of technology. We also performed PCA on the data obtained by ICP-OES and µXRF to assess nutrient content in relation to the accumulation of As”(lines 265-268 of the Revised Version)

The sentence “Indeed, Gallardo et al. [24] proposed….”  was moved in the Introduction (lines 74-80 of the Revised Version)

Line 174 The authors should give the LOD of ICP and µXRF used in the current study for each element.

The LOD of ICP and µXRF has been added in the Methods section (par.2.3-2.4)

Round 2

Reviewer 1 Report

I was sad that reference 11 was not changed, though the authors stated they did.

Author Response

I was sad that reference 11 was not changed, though the authors stated they did.

The comment of the reviewer is right, It is a typo, I’m sorry.  

As regards the previous reviewer’s comment “The equipment used in reference 11 is field portable XRF, not µXRF. Please check other references as well.”

We have selected other references and deleted reference 11

Reviewer 2 Report

The authors have made effort to revise manuscript, but the langguage should be further polished, and novel insights as compared to previous similar studies should be discussed in Results section. I suggest the Conclusion and also Abstract should incluede some key result data.

Author Response

The authors have made effort to revise manuscript, but the langguage should be further polished, and novel insights as compared to previous similar studies should be discussed in Results section. I suggest the Conclusion and also Abstract should incluede some key result data.

The authors have made effort to revise manuscript, but the langguage should be further polished,…

We relied on the MDPI language editing service, as stated in the “English-Editing-Certificate-46924”

….and novel insights as compared to previous similar studies should be discussed in Results section….

Thank you for encouraging us to deepen the discussion of the obtained results. Your comments and suggestions helped us in substantially improving the quality of our manuscript.

We improved the discussion of the results, comparing them with the previuos studies in the literature (Lines 338-350)

….I suggest the Conclusion and also Abstract should incluede some key result data.

Abstract has been improved, by adding the statement “ In conclusion, we demonstrate that this methodological approach based on μXRF analysis is suit-able for monitoring the As and element contents in dried plant tissues without any chemical treatment of samples and that changes in most nutrient concentration can be strictly related to the As content in plant tissue.”  This is to underline the results confirmed in the Table 4, which shows that the variation of nutrients such as Ca, Mn, P, K, is strictly related to As concentration in plant tissues.

Conclusions have been improved, by adding the statement Other methods used for measuring elements in plants, though more sensitive and accurate technologies, are very often slow, laborious and expensive. In conclusion, μXRF-based techniques allow rapidly measuring element concentrations in plants at the lab-scale and to obtain quantitative measurements in near-real time. This approach is also promising for large-scale measurements in plants grown in the field”.